# Molecular Mechanism for Malignant Progression of Gastric Cancer Within the Tumor Microenvironment

**DOI:** 10.3390/ijms252111735

**Published:** 2024-10-31

**Authors:** Tasuku Matsuoka, Masakazu Yashiro

**Affiliations:** 1Department of Molecular Oncology and Therapeutics, Osaka Metropolitan University Graduate School of Medicine, 1-4-3 Asahi-machi, Abeno-ku, Osaka 5458585, Japan; t22738q@omu.ac.jp; 2Institute of Medical Genetics, Osaka Metropolitan University, 1-4-3 Asahi-machi, Abeno-ku, Osaka 5458585, Japan

**Keywords:** gastric cancer, tumor microenvironment, signaling pathway, immune tolerance, microbiota

## Abstract

Gastric cancer (GC) is one of the most common cancers worldwide. Most patients are diagnosed at the progressive stage of GC, and progress in the development of effective anti-GC drugs has been insufficient. The tumor microenvironment (TME) regulates various functions of tumor cells, and interactions between the cellular and molecular components of the TME—e.g., inflammatory cells, fibroblasts, vasculature cells, and innate and adaptive immune cells—promote the aggressiveness of cancer cells and dissemination to distant organs. This review summarizes the roles of various TME cells and molecules in regulating the malignant progression and metastasis of GC. We also address the important roles of signaling pathways in mediating the interaction between cancer cells and the different components of the GC TME. Finally, we discuss the implications of these molecular mechanisms for developing novel and effective therapies targeting molecular and cellular components of the GC TME to control the malignant progression of GC.

## 1. Introduction

The tumor stroma is composed of various cellular and non-cellular factors, such as fibroblasts, vascular endothelial cells, immune cells, cytokines, and the extracellular matrix (ECM), and each of these factors influences the others to create a complex tumor microenvironment (TME) [1]. The molecular mechanisms underlying the involvement of the TME in cancer progression were unknown for many years. However, recent technological innovations such as single-cell analysis technologies and developments in simulation technology based on mathematical science have revealed several interactions between immune cells and stromal cells in tumor tissue. Mechanisms of tumor growth induction through organ-related factors including the bacterial flora have also been elucidated, highlighting the importance of the TME in the aggressiveness, invasion, dissemination, and colonization of cancer cells to secondary foci [2]. In this review, we summarize the current findings on the molecular mechanisms of the involvement of TME in various stages of the malignant progression of gastric cancer (GC). We also address how various signaling pathways interact with the GC TME and contribute to cancer’s progressive and metastatic potential. Finally, we discuss challenges in the targeting of TME components for targeting TME components to improve therapeutic outcomes, and provide future perspectives.

## 2. Component of the TME in GC

The TME of GC has great plasticity and undergoes continuous alterations and stage-specific modifications in response to various cancer cell-intrinsic and -extrinsic factors. Networks of cytokines and growth factors disrupt signaling pathways, and modified molecular signatures in the stroma characterize these changes in the TME. Previous efforts to characterize TME have underscored the pivotal role of the interaction between tumor cells and stroma in driving GC carcinogenesis, progression, and metastasis.

### 2.1. Stromal Cells

#### 2.1.1. Cancer-Associated Fibroblasts (CAFs)

Stromal cells play important roles in tissue homeostasis, disease progression, inflammatory and fibrotic states, and wound healing processes. In neoplasms, a better understanding of the complex landscape of stromal cells could have diagnostic, prognostic, and therapeutic significance [3].

CAFs are a main resource of the stroma and secrete soluble factors such as chemokines, cytokines, and growth factors to employ distinct cell types to the TME and reprogram them [4]. CAFs also produce inflammatory ligands and ECM proteins, which in turn promote tumor growth, invasion, and migration of cancer cells, as well as immune evasion and resistance to anti-cancer therapies [5]. During tumor progression, CAFs promote the secretion of key structural proteins, including collagen type I-V and elastin, which participate in basement membrane (BM) formation [6], inflammation [7], and angiogenesis [8]. Moreover, CAFs produce matrix metalloproteinases (MMPs), which are closely associated with the degradation of the ECM and contribute to ECM remodeling [9].

CAF-specific markers can be utilized to identify biomarkers of GC, and they play important roles in diagnoses, prognoses, and developing novel therapeutic strategies. Biomarkers such as alpha-smooth muscle actin (α-SMA), vimentin, desmin, cadherin 11, integrin α1β1, and MMPs have been used to identify CAF [10]. A comprehensive analysis of multiple patient cohorts showed that GC is one of the cancers in which CAF markers functioned as primary biomarkers and were correlated with worse prognosis [11]. The use of integrated single-cell and bulk RNA sequencing analyses has enabled the identification and validation of a novel nine-gene CAF-related signature (GLT8D1, NRP1, PPP1R26, SERPINE1, TMSB15A, ZFYVE27, AARSD1, GPX3, OLFM3) that can be used to predict the prognosis of GC. Notably, most of these CAF-related genes showed a significant role in malignant progression, indicating that the knowledge obtained in that study could contribute to the development of prognostic biomarkers [12].

Several studies have clarified that CAFs dominate the GC TME and are extensively included in the progression of GC. A recent study revealed that CAFs secrete transforming growth factor (TGF)-β1, which in turn enhances the phosphorylated Smad2/3 pathway in normal fibroblasts, resulting in their transformation into CAFs with a high expression of α-SMA, fibroblast activation protein, and platelet-derived growth factor receptor beta (PDGFRB) [13]. It was later revealed that an amplified secretion of insulin-like growth factor binding protein-7 (IGFBP7) from these CAFs facilitated migration and invasiveness of GC cells and caused stemness. Interleukin (IL)-1α, IL-1β, and tumor necrosis factor (TNF) secreted from diffuse-type GC cells induce very high expression of rhomboid 5 homolog 2 (RHBDF2) in gastric stromal fibroblasts, leading to increasing motility in the ECM and promoting the invasion of diffuse-type GC cells [14]. RHBDF2 upregulates invasion-related genes in CAFs by mediating TGF-β signaling, and CAFs with high RHBDF2 expression induced a lymphatic invasion of GC cells in an in vivo experimental model, indicating a molecular mechanism by which the inflammatory environment in GC is associated with poor survival. Additionally, the secretion of an AXL receptor ligand, GAS6, by gastric CAFs contributes to the viability and epithelial–mesenchymal transition (EMT) of GC cells [15]. Downregulation of AXL suppressed this phenotype and peritoneal metastasis in a GC mouse model. Targeting AXL thus possesses great promise for GC therapeutics. Metastasis is a multi-step and complicated procedure that is induced at the cancer–stroma interference. Among numerous contributive factors, CAFs in the stroma are particularly vigorous in their facilitation of GC metastasis as significant tumor-adjacent stroma factors [13]. CAFs produce ECM-remodeling and -degrading enzymes that substantially alter interactions in the ECM and promote the invasion ability of cancer cells [16]. It has been observed that the lysyl oxidase-like 2 (LOXL2) enzyme, which catalyzes the crosslinking of ECM to construct stable collagen I, was increasingly secreted by gastric CAFs compared with normal gastric fibroblasts. LOX2 activated the FAK/Src pathway, leading to enhanced migration and invasion of GC cells [17]. Similarly, CAFs expressing podoplanin, a transmembrane protein associated with RhoA, increased the invasion capacity of GC cells [18]. A recent study involving a transcriptome analysis of patient cohorts demonstrated that the intratumoral angiogenesis score is associated with the fibroblast fraction and poor prognosis [19]. Gypican-3 is recognized as an oncofetal protein in hepatic neoplasms and yolk sac tumors. Notably, GC with up-regulation of gypican-3 in the CAFs has been shown to exhibit a poorer response to immune checkpoint inhibitors, and the gypican-3 antibody restored the effect of programmed death-ligand 1 (PD-1) blockage therapy in GC and increased the number of tumor-infiltrated CD8^+^ IFN-γ^+^ T cells [20]. Similarly, another study confirmed that, in GC cases, CAFs with high expression of microfibril-associated protein 2 (MEAP2) are associated with poor prognosis and resistance to adjuvant therapy [21]. MRAP2-positive CAFs can induce therapeutic resistance characterized by T cell dysfunction and M2 macrophage polarization. The down-regulation of FERMT2 (fermitin family homolog-2) and NOVA1 (neuro-oncological ventral antigen 1) and the up-regulation of SERPINE2 and IGFBP7 facilitate M2 macrophage polarization, leading to immune evasion [22,23,24,25]. CAFs can also inhibit M1 macrophage polarization with tumor-suppressive activity [26]. It was recently demonstrated that Helicobacter pylori (H. pylori)-induced inflammation activated the NF-κB signaling pathway in GC cells. This activation also drove PIEZO1-YAP1-connective tissue growth factor (CTGF) signaling, resulting in the secretion of CTGF. The CTGF secretion in turn induced the infiltration of α-SMA^+^ CAFs, leading to ECM remodeling that was beneficial to PIEZO1 function [27].

Scirrhous gastric cancer (SGC) represents biological aggressiveness that involves a rapid infiltration into the gastric wall, advancing invasion into the serosa membrane, and seeding into the peritoneum [28]. The presence of a paracrine loop between cancer cells and stromal fibroblasts led to greater aggressiveness of the SGC compared to low metastatic GC. SGC altered the expression profiles of glycolysis-related genes, such as lactate dehydrogenase A (LDHA) and enolase 2 (ENO2), in the CAFs, suggesting that the metabolic switch to glycolysis in CAFs may provide the metastatic capacities of SGC cells [29]. Another study demonstrated that chemokine (C-X-C motif) ligand (CXCL)12 secreted by CAFs enhanced both tumor progression and the invasive phenotype of GC cells in tissue culture by clustering integrin β1 protein [30].

In addition to paracrine signaling, extracellular vesicles (EVs) and exosomes secreted from GC cells are critical in CAF function and recruitment at tumor sites. CAF-derived exosomal miR-29b-1-5p increased vasculogenic mimicry and migration via upregulation of N-cadherin and vimentin through the VSIG1/ZO-1 axis [31]. In another study, CAF-secreted exosomes exhibited higher CD9 expression compared with exosomes from normal fibroblasts. Furthermore, CAF-secreted exosomes are selectively taken up by SGC cells, but not by other types of GC cells. Exosomes from CAFs enhance SGC cell migration and invasion, an effect that can be blocked by antibodies or the siRNA targeting exosomal CD9 [32]. Similarly, a comprehensive proteomic analysis revealed that CAFs-derived EVs containing Annexin A6 promoted resistance to a cisplatin tubular network formation in GC cells by activating a focal adhesion kinase (FAK)/YAP axis via the stabilization of β1 integrin on the surface of cancer cells [33].

As mentioned, growing evidence demonstrates that reprogramming CAFs contributes to chemo-resistance acquisition. It has been shown that GC cells modulated transcriptional alterations of p53 in CAFs compared to normal NFs and had a pivotal role in the secretion of IL-6 by CAFs. An oncolytic adenovirus, OBP-702, induced the wild-type *p53* gene, leading to apoptosis and autophagy in GC cells [34]. The up-regulation of calponin1 (CNN1) in CAFs induced the contraction of CAFs and contributed to stromal sclerosis. Increased matrix stiffness occurred on GC cells, thereby causing 5-fluorouracil resistance following the activation of YAP signaling [35]. Meanwhile, recent studies have revealed that in GC cells, CAFs-derived IL-8 caused cisplatin resistance in GC by activating NF-κB and upregulating the ATP-binding cassette and sub-family B (ABCB1) [36], and it increased the expression of PD-L1 via the NF-κB, p38, and JNK signaling pathways [37].

A comprehensive study has shown that in patients with GC, upregulated levels of circulating CAFs, but not circulating tumor cells (CTCs), are associated with worse survival [38]. In those patients, circulating CAFs and CTCs were clustered, and most of the CTCs showed the characteristics of EMT. The levels of CTCs in the bloodstream are elevated when a patient is treated with a combination of 5-fluorouracil, irinotecan, and oxaliplatin, suggesting that uncovering the interaction between CAFs and the release of CTCs upon chemotherapy may expand our understanding of the mechanisms of metastatic niche construction and chemo-resistance.

Indeed, CAFs may possess critical promise as a latent therapeutic target for GC therapy.

#### 2.1.2. Cancer-Associated Adipocytes (CAAs)

Adipocytes are the principal cellular elements comprising the GC TME, and accumulated evidence suggests that adipocytes induce increased tumor progression via mutual and dynamic interaction between cancer cells and adipocytes [39]. In particular, adipocytes in the TME are transformed into CAAs by signals from cancer cells. These CAAs secrete fatty acids into the adjacent tissues which can be consumed by GC cells [40]. In addition to serving as an energy source, fatty acids play a role in various signaling pathways that can influence cell survival, proliferation, and invasiveness [41]. For instance, our previous study demonstrated that linoleic acid (LA), an important (n-6)- polyunsaturated fatty acid, could increase GC cell invasion and peritoneal metastasis in mice through regulation of the COX-1/extracellular signal-regulated kinase (ERK) 1/2 pathway [42]. LA also enhanced angiogenesis by inhibition of angiostatin via plasminogen activator inhibitor 1 [43]. A recent study revealed that a phenotypic alteration of omental adipose tissue into CAFs induced by the co-culture with GC cells enhanced the malignant progression of GC cells, with increased IL-6 and plasminogen 1 secretion [44]. Likewise, the omentum, which comprises well-vascularities and adipose tissue layers, is a familiar site of peritoneal metastasis. CXCL2 secreted from omental adipocytes promoted the proliferation and invasion of GC cells through AKT activation. In addition, GC cells after omental stimulation induced hypoxia-inducing factor (HIF)1α and vascular endothelial growth factor A (VEGFA) overexpression [45]. Meanwhile, exosomal miR-155 secreted by GC cells has been shown to play a role in repressing lipid metabolism and inducing brown adipose differentiation by targeting C/EPBβ of adipose mesenchymal stem cells, which has an important role in cancer-associated cachexia [46].

In addition, adipocytes can be responsible for chemotherapeutic resistance, since their interaction with fibroblasts leads to the tolerance of anti-cancer therapy by producing ineffective secondary metabolites [47]. Understanding the mechanism of communications between CAAs and GC cells, particularly involving fatty acids, may implicate the direction for developing novel therapies.

### 2.2. Vasculature Cells (Endothelial Cells, Pericytes)

Endothelial cells are important as a constituent part of the cardiovascular system and are essential to angiogenesis and immune response [48]. They modulate the passage of substances via tight cell junctions and line the surfaces of blood vessels. Endothelial cells, along with basal lamina and pericytes, develop the construction of blood vessel walls. Endothelial cells prompt intravasation, which leads to the migration of cancer cells into the blood vessels [48]. Pericytes are cells situated between the BE and endothelial cells of capillaries and serve as multipotent mesenchymal cells which constitute the fundamental structure of capillaries together with endothelial cells and BE [49]. They play an important role in forming angiogenesis, ensuring the structural integrity of the capillary system, regulating blood flow, and closely associating with the maintenance of homeostasis in living cells. Pericytes can differentiate into adipocytes, fibroblasts, smooth muscle cells, osteoblasts, and macrophages [49]. Tumor progression depends on a blood supply, and tumors promote neovascularization by declining the BE of blood vessels during rapid growth [50]. The BE degrades by producing angiogenic factors like PDGF, VEGFA, HIF-1, and MMPs. This BE degradation activates the migration of endothelial cells and pericytes to the tumor area, which leads to TME angiogenesis [50]. NOVA2 is a well-recognized oncogene and is a potent alternative splicing regulator. NOVA2 has been shown to be elevated in cancer endothelial cells of GC, which is closely associated with poor survival and lymph node metastasis [51]. Upregulation of exosomal miR-214-3p in vascular endothelial cells significantly suppressed the expression of ubiquitin editing enzyme A20, which resulted in apatinib resistance to the activity against angiogenesis [52]. Single-cell RNA sequencing (scRNA-seq) data, which were obtained from the Gene Expression Omnibus (GEO) database, identified five distinct tumor endothelial cell (TEC) clusters in GC. The majority of TECs were relevant to worse survival, and HIPPO, NOTCH, and RAS among TEC clusters contributed to GC growth and metastasis. They also constructed eight risk genes (RAB19, KIF24, TCF20, ACLY, SUSD1, ANGPT2, RNASE1, and NPTX1) to develop a risk signature, which could predict the outcome of patients with GC and the response to immunotherapy [53]. Remarkably, GC cells induced pericytes to transform into CAFs through exosome-mediated BMP transfer and activation of the PI3K/AKT and MEK/ERK pathways, thereby inducing the malignant progression of cancer [54]. The role of stroma and vasculature cells in driving GC malignant progression and metastasis is summarized in Figure 1.

### 2.3. Immune Cells

#### 2.3.1. Innate Immune Cells

In the past several decades, immune systems have ascertained roles in host defense against cancers via a variety of mechanisms, leading to the extraordinary progression of cancer immunotherapies. Numerous studies have highlighted the pivotal role of communication between tumor cells and immune cells in promoting GC’s malignant progression and immune tolerance (Figure 2). The innate immune system consists of diverse populations of immune cells, containing macrophages, neutrophils, monocytes, innate lymphoid cells (ILCs), dendritic cells (DCs), and myeloid-derived suppressor cells (MDSCs), which are closely associated with innate immunity against pathogens to maintain homeostasis of the host [55].

Macrophages are a pivotal subset of immune cells in this TME. They can change their plasticity, which serves a broad spectrum of functions, such as tissue homeostasis, removal of pathogens, modulation of inflammatory responses, and tumor progression [56]. It has been shown that the postnatal development of macrophages occurs from circulating monocytes derived from bone marrow hematopoietic stem cells [57]. While macrophage activation is diverse and complex, macrophages can be categorized into classically activated M1 macrophages and alternatively activated M2 macrophages based on the interactions of the cytokines secreted by the CD4^+^ T helper cell subpopulation [58]. M1 macrophages express elevated levels of pro-inflammatory cytokines, and MHCII are referred to as anti-tumor, whereas M2 macrophages, which express greater levels of immunosuppressive factors such as IL-10 and TGF-β, are referred to as pro-tumorigenic. A recent study presented that M2 macrophage polarization, PD-L1 expression, tumor proliferation, and metastasis were repressed by LOC339059. The mechanism acts partly through decreasing the function of the IL-6/STAT3-signaling axis and the transcriptional activation of IL-6 regulated by c-Myc [59]. Notably, a series of bioinformatics analyses detected hub genes of COL1A1, COL4A1, COL12A1, and PDGFRB as significant prognostic biomarkers correlated with M2 macrophage infiltration in GC [60].

Conventionally, it has been described that tumors recruit both tissue-resident macrophages and circulating monocytes to the TME and polarize them to an M2 phenotype, producing tumor-associated macrophages (TAMs) through diverse soluble factors [61]. Meanwhile, accumulating evidence indicates that TAMs comprise not only a homogeneous population, but also an integrated population of macrophages harboring both M1 and M2 phenotypes found in several kinds of solid tumors [62,63]. TAMs have been revealed to promote gene alterations, fibrosis, lymphocyte exclusion, angiogenesis, immune tolerance, migration, and metastasis, resulting in malignant tumor progression. TAMs can induce an inflammatory situation by secreting cytokines [64]. TAMs also suppress antitumor immunity by increasing the expression of immunosuppressive surface proteins, releasing reactive oxygen species, and secreting chemokines that recruit regulatory T lymphocytes (Treg) cells [65]. Additionally, TAMs enhance cancer angiogenesis and metastasis by secretion factors such as VEGF and MMP enzymes that remodel the TME and promote tumor cell migration [66]. In GC, macrophages were identified to secrete CXCL8 in TAMs under hypoxic environments, which phosphorylated the Janus kinase/signal transducer and activator of transcription 1 (JAK/STAT1) pathway through interaction with the CXCR1/2 on the plasma membrane. Consequently, the STAT1 transcription factor induced up-regulation of IL-10 and polarized M2-type macrophages, which resulted in a positive feedback loop between macrophages and GC malignant progression, such as growth and invasion [67]. TAM-derived glial cell-derived neurotrophic factor (GDNF) enhanced liver metastasis of GC through autophagic activity of GDNF family receptor alpha 1 (GFRA1). A significantly high expression of the GDNF protein was found in the invasive margins of the GC liver metastatic niches, which are the frontiers for the interaction between tumor cells and TAMs [68]. Similarly, another study displayed the clinical significance of a potent immunosuppressive macrophage signature, M2_TS_ (MRC1, MS4A4A, CD36, CCL13, CCL18, CCL23, SLC38A6, FGL2, FN1, MAF), in GC patients based on bioinformatics analysis. M2_TS_-expressing macrophages showed a highly immunosuppressive phenotype which was correlated with more persistent tumor activity. An increase in the M2_TS_ macrophage transcriptional signature in the peritoneal fluid was more closely associated with the prognosis of patients with GC compared with conventional cytology [69]. Up-regulation of M2-polarized TAM-derived exosomes, such as miR-21 [70], miR-588 [71], and lncRNA CRNDE [72], results in the cisplatin resistance of GC cells. Meanwhile, miR-223 induces GC cell resistance to doxorubicin [73]. Recently, the importance of M1-like TAM was described, because it enhanced the invasion of liver cancer cells [74]. In GC, M1-like TAMs released CXCL9,10,11, which led to the regulation of efficacy for PD-L1/PD-1 blockades in GC, suggesting that M1-like TAMs are good candidates for anticancer therapy [75].

Neutrophils have been acknowledged as highly abundant innate immune cells in bone marrow as well as peripheral blood [76]. When exposed to exogenic pathogens, activated neutrophils can promptly mobilize to sterilized or infected inflammation sites and function in various ways, such as cytokine release, phagocytosis, and degranulation [77]. Neutrophil extracellular traps (NETs), mainly consisting of granule proteins and chromatin, have demonstrated the involvement of NETs in various non-infectious diseases, including chronic inflammatory conditions, autoimmune diseases, and malignancies [78]. A previous study demonstrated that CD66-positive mature light-density neutrophils (a subpopulation of neutrophils with enhanced capability of producing NETs) were clustering in the peritoneal cavity of patients who underwent laparotomy due to GC. NET presence was found to be related to abdominal recurrence of cancer [79]. Peritoneal metastasis of GC has been found to induce the recruitment of neutrophils and the development of NET. Neutrophils isolated from the ascites of patients with peritoneal metastasis facilitated the growth, invasion, and EMT of GC cells. Bioinformatic analysis based on The Cancer Genome Atlas Program (TCGA) and GEO databases identified LIF, an immunomodulatory signaling molecule, as a potent target for neutrophil infiltration in the TME. Interestingly, the development of NETs and peritoneal metastasis of GC enhanced the TGF-β-Smad-LIF axis [80]. NETs in the TME and peripheral blood were closely associated with lymph node metastases, chemo-resistance, and poorer short-term outcomes in GC patients. NETs also have a higher diagnostic value than CEA and CA19-9 in GC patients as serum biomarkers [81].

DCs are generally acknowledged as persuasive antigen-presenting cells (APCs), which can link innate and adaptive immunity and are critical in maintaining adaptive immune response. DCs are pivotal in initiating and activating anti-tumor T cell responses in tumor-draining lymph nodes and the TME, leading to an antitumor response [82,83]. DCs can be categorized into plasmacytoid DCs (pDCs) and conventional DCs. Recently, the pDCs have attracted increasing attention in GC initiation and progression. The increased number of circulating pDCs was associated with lymph node metastasis and advanced stages in GC patients [84]. A recent study analyzed data from the TCGA Stomach Adenocarcinoma (TCGA-STAD) cohort and the GSE62254 cohort and constructed a prognostic model in GC. The greater risk score showed a significantly lower overall survival time and demonstrated a positive interaction between increased risk score and infiltration of pDCs in GC [85]. It was displayed that within the TME of GC, H. pylori infection could increase the expression of T cell receptor-inducible costimulatory receptor (ICOS) in pDC and Tregs, suggesting that eradicating therapy for H. pylori might serve as an indirect immune therapy for GC [86]. In peripheral blood, the numbers of pDCs, Treg cells, and ICOS^+^ Treg cells were increased in GC patients compared with healthy individuals [84,87]. Notably, pDCs were mostly recognized in peritumor tissue. In contrast, Treg cells and ICOS^+^ Treg cells were identified in tumor tissue, and pDC infiltration positively contributed to ICOS^+^ Treg infiltration into the cancer tissue of patients with GC [87].

ILCs have lately been discovered as innate immune complements of T cells which play a crucial role in numerous human immune-related diseases, such as infection, autoimmunity, and neoplasms [88]. ILCs are categorized into five main subtypes, namely group 1 ILCs, which include both natural killer (NK) cells and ILC1s, ILC2s, ILC3s, and lymphoid tissue-inducer cells, which contribute to immune reactions to various pathogens, including viruses, bacteria, parasites, and transformed cells. Notably, recent studies have disclosed that ILCs contribute to regulating cancer malignant progression; for instance, ILC2s participate in gastric carcinogenesis via their tumorigenic activity, such as the secretion of cytokines, enhancement of chronic inflammation and cancer development, and M2 macrophage polarization [89,90]. Similarly, H. pylori infection stimulates M2 macrophage polarization, which results in the activation of ILC2. Thymic Stromal Lymphopoietin secreted by macrophages is required to facilitate ILC2 [91]. NK cells are members of a family of ILCs. NK cells can distinguish and kill virus-infected and tumor cells, along with secreting numerous cytokines to modulate adaptive immunity [92]. NK cells are one of the most characteristic and practically important cells among the infiltrating immune cells in GC patients. NK cells have been shown to participate in the anti-tumor immune response, for example, the higher number of NK cells in tumorous tissues is closely associated with better prognosis and survival advantage of adjuvant chemotherapy in GC patients [93,94]. NK signature consists of a 12-gene NK cell-associated signature (CXCR4, RDH8, MAGEA11, CYP19A1, SHOX2, GRB14, SLC35E4, NEK5, AKAP5, MSI2, KYNU, PLCL1) and can predict both the prognosis of GC patients and the efficacy of immunotherapy. A nomogram constructed based on the 12-gene NK cell-associated risk signature along with clinical factors could be employed as a useful tool to predict the survival of patients [95].

MDSCs, comprising a population of myeloid progenitor and immature myeloid cells, are characterized by their immunosuppressive effects and protection of cancer cells from host immune attacks [96]. MDSCs are widely distributed in the circulation and tumor sites of cancer patients and participate in the malignant progression of cancer via various mechanisms, including enhancing vascularization and metastatic niche construction. MDSC has become an essential part of tumor immunology, and evidence of the clinical importance of MDSC in GC has emerged [97]. A recent study using scRNA-seq demonstrated that tumor-infiltrating monocytic MDSCs (TI-M-MDSCs) expressed higher levels of genes (IL1B, CCR1, CXCL2, GRINA, IER3) with immunosuppressive functions than other immunosuppressive cell types [98]. The monocytic MDSCs were most highly expressed in GC tissues among the immunosuppressive cell types assessed, which was closely associated with poor prognosis and immune checkpoint inhibitor (ICI) resistance in the GC tumor immune microenvironment (TIME). Mechanistically, serum cytokines IL-6 and IL-8 can promote CD45(+)CD33(low)CD11b(dim) MDSCs to induce arginase I and contribute to CD8^+^ T cell suppression through the PI3K-AKT signaling pathway [99].

#### 2.3.2. Adaptive Immune Cells

The adaptive immune system, which comprises T and B cells, is essential for reacting to and eradicating foreign pathogens [100]. T cells are principal members of the adaptive immune system, and conventional T cells found in the TME consist of NK T cells, CD8^+^, and CD4^+ ^ T cells; CD8^+^ T (cytotoxic T lymphocytes) cells are the main tumor-infiltrating lymphocytes that operate anti-tumor activities. CD4^+^ T cells, mainly including CD4^+^ T-helper (Th) cells and Tregs, are also critical in cancer immunity [79].

Recent scRNA-seq research on tumor specimens and adjacent normal tissues from nine untreated non-metastatic GC patients displayed that CD8^+^ T cells exhibited low expression levels of exhaustion markers, including programmed cell death protein-1 (PDCD1), cytotoxic T-lymphocyte-associated antigen 4 (CTLA4), Hepatitis A virus cellular receptor 2 (HAVCR2), lymphocyte-activation gene 3 (LAG-3), and T cell immunoreceptor with immunoglobulin and immunoreceptor tyrosine-based inhibitory motif domains (TIGIT), which may partly explain the limited benefit of immunotherapy among GC patients [101]. A study using bulk RNA-seq and two independent datasets of scRNA-seq revealed that Cluster of Differentiation 47 (CD47) and galactin-3 were extensively co-expressed in GC cells with peritoneal metastasis and diffuse-type. Inhibition of galactin-3 resulted in increased expression of TNF, IL2, IFNG, Granzyme B^+^CD8^+^ T cells, and Perforin^+^CD8^+^ T cells, which led to a prompted T cell infiltration [102]. Similarly, a study focused on structuring prognostic signatures for GC based on CD8^+^ T cell marker genes via integrated scRNA-seq uncovered eight CD8^+^ T cell feature genes (CXCR4, NPC2, DDX24, ZFP36, TGFB1, PDCD1, NPDC1, and SRI). It generated a novel risk signature for predicting the survival and the effect of immunotherapy in GC patients. GC patients in the high-risk subgroup revealed a superior ratio of MSI-L/MSS, lower immune checkpoint biomarker expression, and lower tumor mutation burden compared with patients in the low-risk group [103]. The existence of Tregs, particularly the Foxp3-expressing subtype, was considered to be related with a worse prognosis due to its function in suppressing CD8^+^ T cells in GC patients [104,105]. TGF-β and IL-10 have been shown to contribute to the induction of Tregs, and TGF-β further prompts Treg differentiation in the GC TME [106]. Another study demonstrated that the immune checkpoint molecules indoleamine 2,3-dioxygenase-1 (IDO-1) is up-regulated in H. pylori-infected gastric mucosa and enhances differentiation of Treg, and also decreases the ratio of Th17 cells versus Treg and the number of Th1 and Th12 cells [107]. Gastric tumor-infiltrating Tregs play a pivotal role in disease progression by mediating immune tolerance. The extent of infiltration of TNF receptor 2 (TNFR2)-positive Tregs increases with GC malignant progression, serving as an independent risk factor for their prognosis for GC. Activating the TNF-α/TNFR2 pathway prompts Treg activity and induces their immune tolerance [108]. Inhibition of TNFR2 suppresses GC progression by reducing CCR8^+^ Treg infiltration, thereby increasing the effect of anti-PD–1 therapy. Tregs have been considered to intricately contribute to the expression of a variety of cell surface markers and immune checkpoint molecules.

B cells, originating from hematopoietic stem cells, comprise the majority of tumor-infiltrating immune cells and have various activities for immune response [109]. Naive B cells enter the primary follicles of lymph nodes and lymphoid tissues, consist of germinal centers, and switch to immunoglobulin. Tumor-infiltrating B lymphocytes (TIBs) can be found in GC and have been shown to inhibit tumor malignant progression by enhancing T cell response, secreting immunoglobulins, and eradicating cancer cells [110]. In a recent study, the IL-10-producing B cell population in the peripheral blood of GC patients was elevated compared with the control individuals. A variety of cytokines, including inflammatory cytokines TNF-α, VEGF, IL-8, and IL-1β, were secreted by a coculture of patient-derived B cells with GC cells, which was associated with a worse survival of GC [111]. It has been reported that, via integrating scRNA-seq with bulk data, CXCR4 expressed by tumor-infiltrating B cells has a poor prognosis, suggesting that CXCR4 can be an attractive therapeutic target for GC [112]. Ectopic accumulation of B and T cells is initially identified as a tertiary lymphoid structure (TLS) during chronic inflammation or cancer, and matured TLSs are characterized by the existence of B cell-rich regions [113]. More recently, a study presented that TLSs-associated B cells induce the secretion of CXCL13 and granzyme B in CXCL13^+^CD103^+^CD8^+^ tissue-resident memory T cells via LTα/TNFR2 signaling, which is induced by mTOR signal-mediated glycolysis. CXCL13^+^CD103^+^CD8^+^ tissue-resident memory T cells within the TLDs predict a better response in patients with GC receiving anti-PD–1 therapy [114].

Table 1 shows the various factors affecting GC malignant progression within the GC TME.

## 3. Signaling Pathways Related with Malignant Progression of GC Within the TME

The intercellular signaling pathway is an intricate network that mediates cell differentiation, growth, migration, invasion, and apoptosis. Cancer progression is often accompanied by changes in one to several key components in the signaling network, which are caused either by the cancer cells or the cancer-adjacent TME, such as stromal cells and immune cells. Various types of carcinogenic and immune cell signaling within the TME interfere with CAFs and immune cells, promoting cancer development, immune evasion, and resistance to therapies [115]. The interaction of signaling pathways with the GC TME is represented in Figure 3. 

The JAK/STAT pathway is a crucial signal transduction network. Cytokines and growth factors bind to the cell surface receptor tyrosine kinase and activate JAK and, thereby, STAT. This results in ectopic STAT translocation to the nucleus, which contributes to diversified biological processes, such as differentiation, growth, apoptosis, and regulation of immune-related genes [116]. In GC, the JAK/STAT pathway can be phosphorylated pathologically and continually. STAT3 can induce EMT. CAFs can produce IL-6 and hepatocyte growth factor (HGF) to enhance IL-6 receptor upregulation in GC cells. Subsequently, IL-6 receptor-α consists of a complex with glycoprotein 130 and stimulates the JAK/STAT3 pathway to promote EMT, resulting in the metastasis of GC [117,118]. As part of the Disintegrin and Metalloproteinase with Thrombospondin motifs (ADAMTS) family, ADAMTS10, secreted by GC cells, can enter the human monocyte cell line along with modulating downstream TXNIP, which is associated with the JAK/STAT/c-MYC pathway [119]. In addition, ADAMTS10 can change reactive oxygen species and suppress the polarization of M2 macrophages. In a recent study, scRNA-seq analysis was carried out to explore the role of up-regulated circulating neutrophils in cancer malignant progression. Ligand–receptor interactions between neutrophils and cancer cells were enriched in the JAK-STAT signaling pathway through KEGG enrichment analysis. The results revealed that tight relations existed among subclusters of circulating neutrophils, M2 macrophages, and cancer cells, and these relations contributed to GC progression during PD-1 blockade [120].

NF-κB is a transcription factor that mediates the expression of genes involved in numerous functions involving cell survival and immune response in the inflammatory nuclear circumstance [121]. Free active NF-κB (p65) translocates to the nucleus to regulate the transcriptional activity through various bridging proteins and signaling kinases [122]. In the GC TME, IL-8 secreted by CAFs can induce PD-L1 expression in cancer cells through activation of the NF-κB, JNK, and P38 signaling pathways, promoting drug resistance in GC cells [13]. Suppression of IL-8 receptor CXCR1/2 significantly decreases the expression of PD-L1 in GC cells, suggesting that CAFs-derived cytokines may enhance the immunosuppressive ligand expression and prompt an immunosuppressive microenvironment in GC. The activation of NF-κB causes the production of the proinflammatory cytokine IL-1β, which results in the physical activation of p65 in GC [123]. Activating p65 produces an inflammatory TME through fibroblast reprogramming. Meanwhile, GC cell-derived glutaminase prompts M2 macrophage polarization, which is closely associated with the acquisition of trastuzumab resistance in human epidermal growth factor receptor 2 (HER2)-positive GC. NF-κB p65 mediates glutaminase expression through the IQ motif containing GTPase activating protein 1 [124].

Notch signaling is crucial for cell destiny and carcinogenesis. Accumulated evidence indicates that notch signaling is implicated in numerous aspects of cancer biology, including metastasis, tumor angiogenesis, stem cell phenotype, and immunosuppression [125]. Increased expression of Notch3 is implicated in the immune evasion of GC, in correlation with reduced infiltration of activated CD8^+^ T cells and high infiltration of Tregs and M2 macrophages, immature DCs, N2 neutrophils, and CD4^+^ T cells in the TME [126,127]. This suggests that Notch3 may have a significant role as a novel prognostic biomarker in GC [127].

The Wnt signaling pathway participates most commonly in embryonic development, self-renewal of tissue, cell polarity, proliferation, and migration [128]. This pathway involves the classical Wnt/β-catenin pathway, the non-classical Wnt/PCP pathway, and the Wnt/Ca^2+^ pathway. Dysregulation of the Wnt/β-catenin pathway was significantly revealed in over half of the GC cells and the surrounding TME. Malignant ascites is an inflammatory and immunosuppressive TME comprising intricate growth factors, cytokines, chemokines, exosomes, and several suspension cells, including tumor mesothelial cells and immune cells. The Wnt/β-catenin signaling pathway is activated after being stimulated by *the* malignant ascites supernatant, leading to gastric peritoneal metastasis [129]. A study using RNA-seq data assessed genes correlated to Wnt signaling and structured three gene-related molecular subtypes to clarify their activities in the GC TIME, thereby constructing a risk prediction model for clinical settings [130]. Wnt-related genes were revealed to be closely related with ECM assembly and remodeling to stimulate the TME. The seven genes (CHRD, BHLHE41, GRP, GPC3, PAX5, S100A2, DKK1) in the model were involved in the regulation of signaling, such as KRAS, TGF-β, and hypoxia pathways, and the risk prediction model contributed to predicting the prognosis [130].

The Hedgehog signaling pathway is evolutionarily conserved and is related to embryonic development, normal tissue repair, EMT, stem cell maintenance, and other processes [131]. A differential gene expression analysis study in the TCGA cohort revealed that Hedgehog pathway genes were potently enriched in the CAF and highly infiltrated GC [26]. Another study has reported that PD-L1 expression is regulated by the Hedgehog signaling induced by H. pylori in GC. Using GC organoids, the expression of PD-L1 induced by the Hedgehog transcriptional effector GLI was mediated by regulating the mTOR pathway [132,133]. The Hedgehog pathway is activated in GC stem cells [134]. The GC stem cells with stimulated Hedgehog signaling had an important role in chemo-resistance in several kinds of agents, such as 5-fluorouracil, platinum, and taxanes [135].

The fibroblast growth factor (FGF)-FGF receptor (FGFR) signaling cascade plays a crucial role in various cellular processes, such as stemness, growth, anti-apoptosis, and chemo-resistance. Targeting FGF or FGFR therapy is an attractive tool to treat GC with FGF and/or FGFR alterations [136]. Paracrine secretion of FGF7 by gastric fibroblasts has an important role in growth, migration, and invasion in FGFR2-expressing GC cells [137,138]. FGF7 is up-regulated by CAFs, which prompts tumor growth in SGC cells, but not in other-type GC cells [137]. The FGFR2 expression by CAFs in SGS cells was reduced under a hypoxic microenvironment, which caused the switches of the driver signal to the SDF1/CXCR4 signaling pathway [139].

Hippo signaling is an evolutionarily conserved pathway regulating organ size by controlling cell proliferation, apoptosis, and stem cell self-renewal [140]. The central Hippo pathway in organisms is composed of a kinase cascade, MST1/2, and LATS1/2, as well as downstream transcriptional coactivators YAP and TAZ. Dysregulation of the Hippo pathway contributes to cancer malignant progression, including GC [140]. High LATS1/2 expressions are closely associated with worse survival in patients with the microsatellite-stable GC [141]. Furthermore, LATS1/2 function repressed the anticancer immunity of CD8^+^ T cells and immunosuppressive activity in FOXP3^+^ Treg cells. A recent study demonstrated via scRNA-Seq that stimulating Yap/Taz in neutrophils can facilitate the differentiation into CD54^+^ neutrophils and increase their anticancer immunity [142]. The YAP signature genes were detected in CD44^−^CXCR2^−^ neutrophils, and activating YAP could promote the differentiation into CD54^+^ neutrophils and enhance their antitumor activity.

The PI3K/AKT signaling pathway is the most commonly activated in cancers and promotes the growth, survival, and, particularly, the metabolism of cancer cells [143]. Previous studies have shown that CAFs phosphorylate PI3K/AKT signaling pathways, thereby enhancing the construction of vasculogenic mimicry channels [144]. The PI3K/AKT/mTOR pathway mediates the resistance of GC cells to 5-fluorouracil. In 5-fluorouracil-resistant GC cells, CXCL5 derived from TAMs induces the PI3K/AKT/mTOR pathway, leading to enhanced polarization of M2 macrophages and forming the feedback loop [145]. Interestingly, the PI3K/AKT signaling pathway has been considered to play a critical role in immunoregulatory cells, including MDSCs, TAMs, and Tregs, and to substantially participate in macrophage polarization, suggesting that therapeutic interventions targeting PI3K would be available [146].

PDGF/PDGFR and HGF signaling are acknowledged to be closely associated with CAF growth and tumor malignant progression, and the prognostic usefulness of these ligands has been established in several types of cancer, including GC [147,148]. A recent study showed that PDGFRB was closely associated with immune cell infiltration, particularly M2 macrophage infiltration in GC, by using integrated bioinformatics. Up-regulated expression of PDGFRB regulated angiogenesis and the TME, leading to poor prognosis in GC [149]. HGF derived from CAFs in the GC TME prompted the growth and invasion of GC cells by stimulating the HGF/Met/STAT3/Twist1 pathway [150]. GC cells can stimulate hepatic stellate cells (HSCs) within the TME of liver metastases by producing PDGFB. HGF secreted by activated HSCs activates the MET receptor, thus facilitating invadopodia construction and promoting the metastasis of GC cells, indicating that the cooperation of GC cells and HSCs by HGF/MET signaling may be implicated in therapeutic targets for GC liver metastasis [150].

TGF-β is crucial to the homeostasis of epithelial cells and stromal elements and also contributes to oncogenesis and cancer development, acting as both a suppressor and a promoter [151]. A recent study revealed that the activation of TGFBR3-regulated TGF-β signaling was prompted by an extracellular sulfatase, human sulfatase 1, secreted by CAFs. This interaction also causes the promotion of EMT, resistance to chemotherapy in GC cells, and metastasis mediated by TGF-β signaling [152]. Further reviews of the current understanding of the role of TGF-β in the GC TME are accessible elsewhere [153].

The colony-stimulating factor 1 receptor (CFS-1R) is a class III transmembrane tyrosine kinase receptor crucial to monocyte differentiation and progression [154]. Its upregulation is correlated with positive tumors characterized by an immunosuppressive microenvironment. CSF-1R ligands, IL-34 and M-CSF, are secreted by various cells in the TME, implying a pivotal role for the ligands in the TME crosstalk. The clinical importance of CSF-1R was recently reported in GC [155]. According to this study, CSF-1/CSF-1R expression correlated with an advanced stage and metastasis of patients with GC. Moreover, the CSF-1/CSF-1R axis components are extensively expressed in GC and can promote the proliferation, migration, and resistance to anoikis in GC cell lines. The CSF-1R axis is positively correlated with VEGF-A expression, which contributes to tumor angiogenesis. Similarly, another study based on bioinformatics analysis also revealed that CSF-1R was significantly related to malignant progression, immune-active status in TME, and the survival of GC patients [156]. These findings implied that CSF1R might be a potential biomarker for GC diagnosis and prognosis.

The endoplasmic reticulum (ER) is a critical organelle, depositing the greater part of calcium and regulating protein translation. It has an important role in preserving homeostasis in all ER components. The ER stress sensor pathways, involving IRE1/sXBP1, PERK/EIf2α, and ATF6, are activated in response to ER stress [157]. Anterior gradient homolog 2 (AGR2) belongs to the protein disulfide isomerase family and serves as an ER retention protein [158]. It has been demonstrated that the extracellular AGR2 protein was shown to be up-regulated in SGC cells and was closely correlated with the capacity to prompt stromal fibroblasts and stimulate invasion, as well as the resistance to oxidative and hypoxic stresses [159].

In the CXCL12/CXCR4 chemokine axis, both the receptor and ligand are broadly expressed in mammalian cells. It can interact with cancer cells as well as stroma to construct a more comprehensive oncogenic signaling network and contribute to cancer angiogenesis and immunosuppressive activity [160]. In GC, increased CXCL12 expression facilitates invasion and EMT [161]. Meanwhile, the CXCL6/CXCL8-CXCR1 pathway can mediate the motility of tumor-activated neutrophils in GC cells [162].

Immunosuppressive checkpoint molecules, including PD-1, cytotoxic T-lymphocyte (associated) antigen-4 (CTLA-4), T cell immunoglobulin, mucin-domain-containing-3 (TIM-3), LAG-3, and TIGIT, are regularly expressed on T cells and attach to their ligands on other cells, leading to the induction of suppressive regulations on the immune signaling pathway [115]. PD-1 and its ligand, PD-L1, can be expressed on the cell surfaces of cancer, whereas PD-1 is principally expressed on the surfaces of immune cells. The PD-L1/PD-1 pathway functions to protect immune tolerance and allows cancer cells to evade the immune system. In GC, PD-1/PD-L1 expression is closely relevant to immunotolerance and worse survival [163]. Genetic alteration of CTLA-4 in mammalians is considered to be related to GC malignant progression [164]. A recent study has shown that up-regulation of CTLA-4 and PD-L1 was an independent prognostic indicator in patients with GC [165]. TIM-3, induced on tumor-infiltrating T cells and NK cells, is an independent factor involved in worse outcomes in GC patients and has a pivotal role in the malignant progression and metastasis of GC [166,167]. Up-regulation of the TIM-3 ligand galectin-9 has been correlated with TNM stage and vessel invasion in GC [168]. Enhanced expression of TIGIT in the TME has been revealed in patients with GC, along with increased expression of its ligands, CD155 and CD112, which is correlated with immune evasion caused by CD8^+^ T cell repression [169]. The TIGIT-expressing peripheral serum CD8^+^ T cells from patients with GC impaired cellular metabolism and diminished cell activity through CD155/TIGIT signaling, which was reversed by the inhibition of CD155 [170].

## 4. Role of Gut Microbiota in GC TME

The microbiota involves a distinct array of viruses, bacteria, fungi, and protozoa cooperatively forming an intricate environment. These microbial communities, residing in various body habitations, including the oral cavity, gut, lung, skin, and genitourinary tract, have vigorously contributed to energy homeostasis, metabolism, immunologic activity, and gut epithelial health [171]. The accumulating evidence especially demonstrates the function of the gut microbiota in promoting oncogenesis and malignant progression, arranging the complex landscape of TME, and clarifying novel ways to employ the microbiota to regulate anti-tumor immunity [172]. Previously, the gastric microbiota has attracted relatively restricted attention because of the comparatively lower biomass [173]. However, recent studies have demonstrated vigorous evidence to elucidate the relationship between gastric microbiota and GC’s carcinogenesis and malignant progression based on applying advanced next-generation sequencing (NGS) technologies [174]. A previous study has described that tumor specimens of GC obtained from individuals unveiled notably higher levels of five oral-derived species, including Peptostreptococcus stomatis, Streptococcus anginosus, Parvimonas micra, Lackia exigua, and Dialister pneumosintes, which could be adopted as biomarkers to distinguish GC from superficial gastritis. Furthermore, this study also implied that H. pylori infection leads to significant modifications to microbial composition in the stomach [175]. However, the specific alterations in bacterial composition diverge according to the experiments [176,177]. H. pylori can significantly prompt GC oncogenesis by utilizing a variety of toxic factors [178]. H. pylori infection has been shown to induce a meaningful increase in STAT3 activity concurrently with the promotion of FGFR4 in GC cells. STAT3 connected directly to the FGFR4 promoter and formed a positive feedback loop, resulting in the enhancement of cell survival to produce tumorigenic cells. These findings suggest the important role of the gastric microbiota in the processes of gastric carcinogenesis [179]. With the emergence of high-throughput sequencing technology in microbiology, acid-resistant bacteria, except for H. pylori, have also contributed to gastric carcinogenesis. The bacterial genera mostly described to highly exist in GC patients involve Clostridium, Lactobacillus, Streptococcus, Veillonella, Fusobacterium, Prevotella, Lachnospiraceae, and Leptotrichia [178]. For instance, the Streptococcus anginosus lipoprotein TMPC and ANXA2 promote Streptococcus anginosus-induced mitogen-activated protein kinase (MAPK) activation and enhance gastric tumorigenesis through direct connections with gastric epithelial cells [180]. Nevertheless, the mechanisms of how these microbe–host interactions induce GC oncogenesis are still unknown. Gut microbial metabolites can also regulate carcinogenesis through various mechanisms, involving disordering the balance between pro- and anti-inflammatory signaling and constructing a practical intricate of biomolecules [181]. A study investigating the interaction between gastric microbiota and metabolites in GC demonstrated that the metabolome profiles of the tissue samples of GC were potently modulated by Helicobacter *and* Lactobacillus. Discriminative metabolites were found in the amino acid class, carbohydrates, and carbohydrate conjugates. Nucleosides, adenosine, and glycerophospholipids revealed increased relative abundance in the tumor tissues than in the non-tumor tissues, leading to the malignant progression of GC [182]. Moreover, the concentration of metabolites including glutathione, S-adenosylhomocysteine, S-adenosylmethionine, L-cystathionine, and S-methyl-5′-thioadenosine was significantly increased in the GC tumors by KEGG enrichment analysis.

The microbial prototype serves as a critical component that potently forms the activity, plasticity, recruitment, and total activity of the innate and adaptive immune system in the GC TME [178]. A previous study demonstrated that increased expression of hepatoma-derived growth factor (HDGF) and CXCL8 (IL-8) by H. pylori infection induces neutrophil recruitment, which is closely associated with gastric carcinogenesis [183]. In the GC TME, Stenotrophomonas and Selenomonas are positively associated with BDCA2+ plasmacytoid DCs and Foxp3+s Tregs, which are involved in the immune evasion of GC cells and contribute to the GC development [184]. The abundance of the intratumoral Methylobacterium species can suppress the expression of TGFβ and CD8^+^ tissue-resident memory T cells in the GC TME, thereby promoting tumor progression and considerably correlating with a worse prognosis of GC, implying that the gastric microbiota might be involved in regulating TIME positively [91]. A study revealed that patients with GC who carried out Roux-en-Y reconstruction for radical gastrectomy exhibited an elevated level of butyrate to suppress the function of macrophages by modulating the NLRP3 pathway and decreasing the secretion of short-chain fatty acids, suggesting that short-chain fatty acids have a significant role in the prevention of colitis after GC surgery [185].

Based on these studies of gut microbiota, potential implications in diagnostic, predictive, and prognostic biomarkers, as well as new therapeutic targets of GC, could be explored.

## 5. TME-Based Cancer Therapy in GC

So far, the main therapeutic approaches for GC are surgical procedures and chemotherapy, and molecular targeted therapy has also become an established strategy for GC therapy. Conventionally, treatment approaches for GC mainly focused on killing tumor cells, and in the process have elucidated much of the intricate milieu of the TME [186]. Therapeutic strategies targeting CAF signaling pathways within the TME have appeared to be an effective approach to achieving GC reduction. Targeting CAFs has been considered to improve the prognosis of GC patients by inhibiting the tyrosine kinase inhibitor resistance caused by CAFs. Thus, it is crucial to detect potential therapeutic targets that inhibit the function of CAFs. A recent study displayed that treatment of GC cells with the vitamin D receptor (VDR) ligand calcipotriol could very effectively abolish the oxaliplatin resistance of GC cells, which is mediated by inhibiting the tumor-supporting activity of CAF, suggesting that vitamin D may act as a defensive agent in GC chemotherapy [187]. Unfortunately, the discovery of drugs targeting CAFs is challenging at present due to the insufficient understanding of the source, subtype, and function of CAFs and their interaction with GC cells.

Immunotherapies targeting PD-1 and CTLA-4 have applied remarkable clinical utility and greatly informed the study of cancer immunity. Although treatment combining nivolumab with ipilimumab in the phase 3 CheckMate 649 trial did not result in prolonged overall survival compared to chemotherapy alone [188], a recent analysis investigating biomarkers from the CheckMate 649 trial revealed that treatment with these antibodies has possible benefits in Treg-enriched patients [189]. Numerous clinical trials targeting PD-1 and CTLA-4 are under investigation (Table 2). It is essential to identify therapies that can impair or halt immunosuppression and T cell failure, as well as promote TME effector activity. It is particularly crucial to advance the evaluations of combination therapies to reduce the accumulation of immunosuppressive cells, such as TAMs and FOXP3+ Tregs, in the TME while enhancing the functions of CD4^+^ and CD8^+^ effector T cells therein [190]. Such treatments could efficiently remodel the TME and enhance the anti-tumor immune response. A recent study demonstrated that a synergic expression of galectin-3 with CD47 in peritoneal metastatic cells is correlated with diffuse-type GC and tumor relapse. The concurrent administration of galectin-3 and CD47 extensively promoted the remodeling of TAMs and phagocytosis, resulting in an increased T cell response and suppressed tumor growth in an in vivo experimental peritoneal metastasis model [102]. Another research group observed that outstanding efficiency was achieved with the use of SIRPα-Fc in anti-CD47 therapy in human xenograft or patient-derived xenograft models of GC. An ongoing phase 2/3 trial is vigorously investigating the potential of SIRPα/FC fusion protein antibody, evorpacept (ALX148), in combination with trastuzumab, ramucirumab, and paclitaxel for patients diagnosed with metastatic HER2-overexpressing gastric/GEJ adenocarcinoma (NCT05002127). Moreover, a therapy targeting both CD47 and VEGF, along with the administration of a bispecific fusion protein SIRPα-VEGFR1, achieved remarkable enhancements in the TIME and a substantial augmentation of the anti-tumor response, suggesting the substantial benefit of combining anti-CD47 therapy with anti-angiogenic treatment for combatting GC [191].

Chimeric antigen receptor (CAR) T cell therapy is a comparatively novel approach to GC and is becoming more widely administered in cancer immunotherapy [192]. CAR T cells are a subset of genetically engineered T cells that express synthetic receptors designed to recognize specific antigens, and their activity is independent of major histocompatibility complex interactions, allowing them to stimulate the immune system to detect and eradicate cancer cells. Notably, CAR T cell therapy has provided a potential approach to overcome the drug resistance that can occur in patients with advanced GC. Several promising targets for GC therapy have been identified, including carcinoembryonic antigen, claudin 18.2 (CLDN18.2), mesothelin, HER2, and CDH17. Among them, CDH17 CAR T cells and CLDN18.2 CAR T cells have shown good results. Clinical trials including CLDN18.2 CAR T cells that were designed for patients with positive CLDN18.2 expression revealed a significant antitumor response (Table 2). Developments in CDH17 CAR T cell treatments have indicated that this novel immunotherapeutic approach could be a feasible and useful therapy tool for GC. However, sufficient therapeutic efficacy was not observed in all patients [193]. Similarly, chimeric antigen receptor NK cells (CAR-NK) and macrophages (CAR-Mϕ) may serve as breakthrough treatments for GC [194,195]. 

For the past several years, the gastric microbiota has been illustrated to be closely correlated with various GC treatments. A recent study demonstrated that oral administration of Clostridium butyricum inhibited inflammation and immunity, restored intestinal microbiota eubiosis, and thus reduced the incidence of postoperative gastrointestinal complications in GC patients after gastrectomy [196]. Similarly, a clinical study investigating the efficacy of probiotic compounds on postoperative recovery in patients with GC revealed that a shortened recovery of gastrointestinal function was achieved in patients in the probiotic group compared with those in the untreated group [197]. The gut microbiota may be a novel predictive target for GC chemotherapy and immunotherapy. Interestingly, sodium butyrate prompts the effects of cisplatin by increasing the apoptosis of GC cells by directly interfering with the mitochondrial apoptosis-related pathway [198]. In a study investigating the association of gut microbes with GC therapy, patients suffering from HER-2-negative GC with a higher abundance of Lactobacillus in the gut were prone to showing increased microbiome diversity and were related with better responses to anti-PD-1/PD-L1 immunotherapy [199]. Prior antibiotic administration induces gut microbial dysbiosis and alters systemic immune responses, thus impairing the efficacy of PD-1 blockade. This suggests that antibiotic administration may be an actionable target for patients planning to be treated with anti-PD-1 immunotherapy. Nevertheless, the precise impact and mechanism of the gut microbiota on the usefulness of therapy for GC need to be further elucidated.

## 6. Discussion and Future Perspective

GC is the fifth most common malignancy in the world and the third leading cause of cancer-related mortality worldwide [200]. Conventional therapies, including surgery, radiotherapy, chemotherapy, and targeted therapy for GC, have been confirmed to induce the regression of most primary GCs, but these therapies have not helped all patients with GC, and post-therapy recurrence and drug resistance remain a challenging problem [201,202]. As described in this review, during the progression of GC, the crosstalk with the TME potentiates different aspects of tumor metastasis, including growth, invasion, EMT, and angiogenesis, and mediates drug resistance and immune tolerance. The existing research findings regarding the interaction between the TME’s components and GC cells have significantly enhanced our understanding of their critical roles of the TME in cancer growth and metastasis. Single-layer models have proven to be inadequate for clarifying the precise mechanism underlying the complicated network between cancer intricacy and direct pathobiology relevance. It is thus necessary to construct progressive GC models with an appropriate TME, and it is important to recapitulate the TME to further understand these interactions, which closely represent the patient-derived heterogeneity and clarify the cellular interactions in the TME [203]. Under such circumstances, cancer cells yield more accurate oxygen and nutrient gradients. 

Spheroids were developed as a significant in vitro model for the screening of cancer drugs, given their capacity to replicate the principal characteristics of solid tumors—e.g., gene expression, cellular heterogeneity, cell–cell signaling, and drug resistance [204]. In the context of GC, it has been noted that 3D cell spheroids generated using tumor tissues derived from GC patients maintain the histological features, gene expression patterns, and similar phenotypes of the parental source tissues. The 3D cell spheroids can therefore be utilized as a reliable tool for predicting drug sensitivity and toxicity [205]. A recent study constructed an in vitro GC 3D co-culture spheroids model using GC cells and CAFs derived from GC [206]. Combining docetaxel with lovastatin effectively suppressed spheroid growth and promoted apoptosis. The monitoring of the treatment response of 3D tumor spheroids was shown to be an extremely useful and reliable approach for quantifying the overall cell toxicity [206]. The application of a bicellular spheroids model composed of GC cells and CAFs revealed that the administration of an SRC-family kinase inhibitor, dasatinib, resulted in CAFs-induced spheroid compactness, while its effect vanished in monocellular spheroids, conceivably via the inhibition of the expression of CTGF [207].

Organoids are 3D cultures of living tissues that have been used in an ex vivo environment and resemble the extracellular niche in vivo. Organoids can efficiently present the features of morphological constructions, activities, and genetic phenotypes of primary organs [208]. Tumor organoid models can reveal the majority of the biological features of tumor cells in vitro, leading to more precise biological models for basic tumor research. A recent study established an autologous cancer patient-derived organoids/immune cell co-culture system [133]. H. pylori-induced PD-L1 expression is modulated by the Hedgehog signaling pathway in patients with GC. In an investigation using organoid technology, immunosuppressive MDSCs expressing arginase 1, CEA cell adhesion molecule 8, V-domain Ig suppressor of T cell activation, and indoleamine 2,3-dioxygenase 1 were identified in GC tissues. Although nivolumab resistance was observed in GC organoid/immune cell co-cultures, after the removal of MDSCs, the organoids converted sensitively to a PD-1/PD-L1-inhibitor [133]. Such research using 3D methods may offer potentially beneficial insights into the mechanisms underlying the interaction between GC cells and the TME.

Cancer immunotherapy intends to activate the immune cells for cancer killing. Although targeting T cells is optimal immunotherapy, it has several limitations, such as tumor antigen-specific inhibitory mechanisms regulated by Tregs and MDSCs [186]. Therefore, new immunotherapeutic strategies are required. Recently, a biologically therapeutic approach to opsonize cancer cells and prompt their death by immune effector cell targeting has received extensive attention. NKG2D (natural killer, group 2, member D) is an activated receptor that is expressed in T cells, γδT cells, and NK cells [209]. NKG2D, a class of C-type lectin-like protein receptors of the CD94/NKG2 superfamily, regulates innate and adaptive immune responses. Combined with its ligands, MHC class I chain-related proteins A and B (MIC-A/B) and UL-16 binding proteins (ULBPs) trigger the immune anti-tumor effect, including NK cell-mediated recognition and elimination of cancer cells [210]. Accumulating evidence has shown that NKG2D is an important molecule introducing an immune response underlying the malignant progression of GC. For instance, H. pylori infection causes proteolytic shedding of NKG2D ligands in gastric epithelial cells, suggesting that evasion of the NKG2D system suppresses immune surveillance, initiating GC [211]. H. pylori infection also produces sMICA/B, and its secretion is regulated by metalloproteases closely associated with MICA/B shedding. A recent study revealed that inhibition of DKK1, an antagonist of the classical WNT pathway, upregulated NKG2D and improved the immune-activating and tumor-killing ability of NKG2D-chimeric antigen receptor-T (NKG2D-CAR-T) cells. Moreover, an increased antitumor effect was demonstrated by combining DKK blockade with NKG2D-CAR-T cell therapy compared with a single treatment, which could provide new insight into GC immunotherapy [212] Meanwhile, NKG2D ligands can be cleaved by a disulfide isomerase and several proteases belonging to a disintegrin and metalloproteinase (ADAM) and MMP families [213,214,215]. Loss of recognition of cancer cells by NKG2D receptors causes immune evasion of tumor cells from NK cell cytotoxic activity. Previous research has demonstrated that ADAM10 has shedding effects on various NKG2D ligands in Hodgkin’s lymphoma cell models, and the ADAM10 inhibitor revealed a significant ability to reduce the shedding of the NKG2D ligands [216]. In the case of GC, MMP induced cleavage of NKG2D ligands in GC cells, which resulted in reduced NK cell immune surveillance. Inhibiting MMP restored the expression of the NKG2D ligand, consequently promoting NK activity in GC cells, suggesting that the use of MMP inhibitors is a significant strategy to promote NK-cell-mediated immunotherapy for GC [217]. Another activating receptor in NK- and T cell-modulating killing of cancer cells is DNAX accessory molecule-1 (DNAM-1), a transmembrane glycoprotein expressed by NK cells, T cells, and macrophages. DNAM-1 ligands were identified as nectin-2 (CD112) and the poliovirus receptor (CD155), which were included in the nectin/nectin-like family [218]. A recent study has shown that in GC cell lines, AGS expressed a higher density of CD112 and CD155 that can induce DNAM-1 to activate NK cells, leading to sensitivity to killing by NK cells [219].

The challenge in the treatment of GC is the identification of patients who would benefit from immunotherapy. It appears that combined anti-cancer treatments with TME-targeted agents are the most useful and may lead to personalized therapies. Overcoming acquired drug resistance in the TME is also an important obstacle. These challenges are complicated by the TME’s capacity to be positively involved in tumorigenesis. The rearrangement of dysfunctional TMEs may soon provide a significant advantage for modulating and mitigating cancer, assisted by the pivotal achievement in cancer immunotherapy reported thus far. Moreover, the emergence of knowledge about the human body’s immunometabolism is considered pivotal to advancing new immunotherapies and overcoming resistance against conventional and ICI therapies [220]. The recent advances in bioinformatics analyses, such as NGS, seRNA-seq, and metabolomics, can be applied to detect novel GC biomarkers, practical phenotypes in TME, drug resistance, and novel targeted immunotherapies [221]. Such research will result in improved GC diagnoses and the improvement of revolutionized TME cell-specific targeted therapies with fewer advertise events, thereby improving the prognoses of patients with GC. Several promising ongoing clinical trials targeting the GC TME may contribute to applying these cells in GC therapies (Table 2). These tools are expected to play a critical role in enriching TME-targeting therapy with patient subgroups most likely to benefit from treatment.

## 7. Conclusions

The TME is a highly complex and dynamic milieu, with various tumor cells, stromal cells, vasculature cells, and immune cells supporting tumor malignant progression by upregulating signaling pathways correlated with angiogenesis, invasion, and immune tolerance. Because the GC TME is crucial in regulating the process of cancer progression and metastasis, it is also a highly practical candidate for therapeutic interventions. Furthermore, the gut microbiota plays a pivotal role in promoting malignant progression and in modulating TME heterogeneity through its immunomodulatory capacities. Future unraveling of the cellular and molecular interrelationships in the GC TME is expected to lead to a more comprehensive knowledge of the tumorigenesis process. Such knowledge could lead to improvements in the efficacy of currently existing therapeutic approaches, and, thus, to enhanced patient outcomes. 

## Figures and Tables

**Figure 1 ijms-25-11735-f001:**
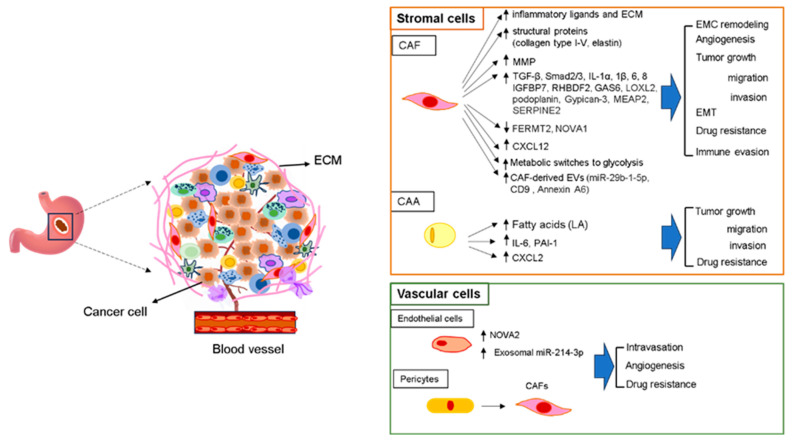
Representative image of the formation of TME (1). Construction of the tumor microenvironment and the detailed processes involved in recruiting stromal and vasculature cells are shown (see text for details). ↓ = downregulation; ↑ = upregulation.

**Figure 2 ijms-25-11735-f002:**
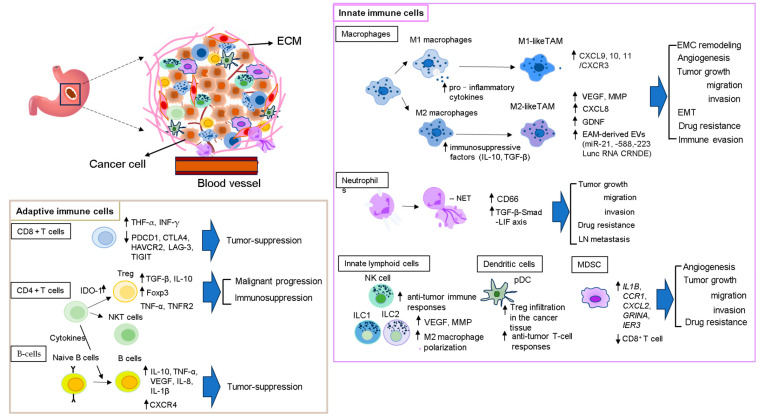
Representative image of the formation of TME (2). The construction of the tumor microenvironment and the detailed processes involved in recruiting immune cells are shown (see text for details). ↓ = downregulation; ↑ = upregulation.

**Figure 3 ijms-25-11735-f003:**
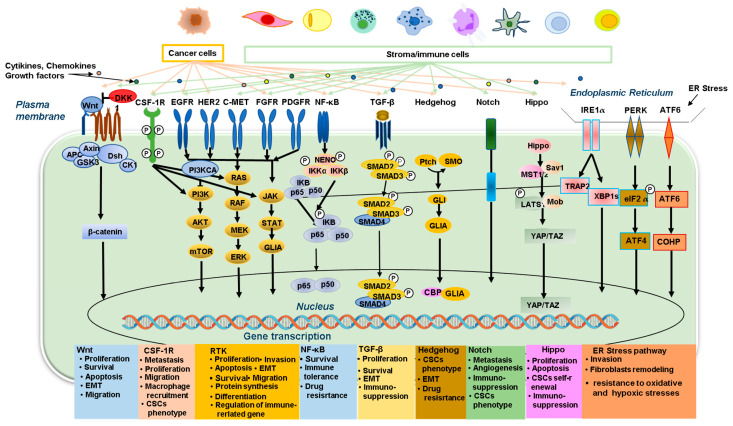
Schematic representation of the main signaling pathways that are activated in tumor cells or tumor-associated stromal cells and modulate the interplay within the TME, promoting tumor development and metastasis (see text for details).

**Table 1 ijms-25-11735-t001:** The summary of TME components affecting the malignant progression of gastric cancer. TME = tumor microenvironment, GC = gastric cancer, CAFs = cancer-associated fibroblasts, CAAs = cancer-associated adipocytes, TAMs = tumor-associated macrophages, DCs = dendritic cells, MDSCs = myeloid-derived suppressor cells, Treg = regulatory T cell, TGF = transforming growth factor, IGFBP7 = insulin-like growth factor binding protein-7, RHBDF2 = Rhomboid 5 homolog 2, ECM = extra cellular matrix, EMT = epithelial–mesenchymal transition, LOXL2 = lysyl oxidase-like 2, MEAP2 = microfibril-associated protein 2, FERMT2 = fermitin family homolog-2, NOVA = neuro-oncological ventral antigen, CTGF = connective tissue growth factor, LDHA = lactate dehydrogenase A, ENO2 = enolase 2, SGC = scirrhous gastric cancer, CXCL = chemokine (C-X-C motif) ligand, FAK = focal adhesion kinase, CNN1 = calponin1, IL—interleukin, LA = linoleic acid, ERK = extracellular signal-regulated kinase, JAK = Janus kinase, STAT1 = signal transducer and activator of transcription, GDNF = glial cell-derived neurotrophic factor, NETs = neutrophil extracellular traps, pDCs = plasmacytoid DCs, ICOS = inducible costimulatory receptor, NK cells = natural killer cells, PDCD1 = programmed cell death protein-1, CTLA4 = T-lymphocyte-associated antigen 4, HAVCR2 = Hepatitis A virus cellular receptor 2, LAG-3 = lymphocyte-activation gene 3, TIGIT = T cell immunoreceptor with immunoglobulin and immunoreceptor tyrosine-based inhibitory motif domains, CD47 = cluster of differentiation, IDO-1 = indoleamine 2,3-dioxygenase-1, TNFR2 = TNF receptor 2, VEGF = vascular endothelial growth factor, TLS = tertiary lymphoid structure, ↓ = downregulation; ↑ = upregulation, N.A. = not applicable.

TME Components	Specific Elements	Alterations	Comments	Ref.
Stromal cells				
CAFs	GLT8D1, NRP1, PPP1R26, SERPINE1, TMSB15A, ZFYVE27, AARSD1, GPX3, OLFM3	↑	A novel 9-gene CAF-related signature predicts the prognosis of GC, and most of these CAF-related genes had a significant role in malignant progression.	[12]
	TGF-β1, Smad2/3 pathway IGFBP7	↑	*The Smad2/3* pathway transforms normal fibroblasts into CAFs. IGFBP7 facilitated GC cells’ migration and invasion properties and caused stemness.	[13]
	RHBDF2	↑	RHBDF2 expressed in gastric stromal fibroblast increases motility in ECM and promotes diffuse GC cell invasion.	[14]
	GAS6	↑	GAS6 secreted by gastric CAFs contributes to the induction of GC’s viability and EMT.	[15]
	LOXL2	↑	LOX2 secreted by gastric CAFs activated the FAK/Src pathway and enhanced migration and invasion of GC cells.	[17]
	Podoplanin	↑	Podoplanin secreted by gastric CAFs and associated with RhoA increased the invasion capacity of GC cells.	[18]
	Gypican-3	↑	Gypican-3 in the CAFs contributed to a poorer response to immune checkpoint inhibitors and increased the tumor-infiltrated CD8^+^ IFN-γ^+^ T cells.	[20]
	MEAP2	↑	MRAP2-positive CAFs can cause therapeutic resistance characterized by T cell dysfunction and M2 macrophage polarization.	[21]
	FERMT2, NOVA1 SERPINE2, IGFBP7	↓ ↑	These proteins are closely associated with M2 macrophage polarization and immune evasion.	[22,23,24,25]
	CTGF	↑	CTGF secretion via PIEZO1-YAP1-CTGF signaling induced the infiltration of α-SMA^+^ CAFs, leading to ECM remodeling.	[27]
	LDHA, ENO2	↑	The metabolic switch to glycolysis in CAFs may provide the metastatic capacities of SGC cells.	[29]
	CXCL12	↑	CXCL12 in CAFs enhanced tumor progression and invasive phenotype of GC cells by clustering integrin β1 protein.	[30]
	Exosomal miR-29b-1-5p	↑	miR-29b-1-5p increased Vasculogenic Mimicry and migration by upregulating N-cadherin and vimentin through the VSIG1/ZO-1 axis.	[31]
	CD9	↑	Exosomal CD9 from CAFs enhances SGC cell migration and invasion.	[32]
	Annexin A6	↑	Annexin A6 promoted resistance to cisplatin by activation of the FAK/YAP axis via the stabilization of β1 integrin.	[33]
	OBP-702	↑	OBP-702 induced the wild-type *p53* gene, which led to induced apoptosis and autophagy in GC cells.	[34]
	CNN1	↑	CNN1 in CAFs induced the contraction of CAF and contributed to stromal sclerosis, causing 5-fluorouracil resistance via YAP signaling.	[35]
	IL-8	↑	IL-8 caused cisplatin resistance by activating NF-κB and increased PD-L1 expression via the NF-κB, p38, and JNK in GC cells.	[36,37]
CAAs	LA	N.A.	LA increases GC cell invasion and peritoneal metastasis in mice through the regulation of the COX-1ERK 1/2 pathway.	[42]
	IL-6, plasminogen 1	↑	A phenotypic alteration of omental adipose tissue into CAFs enhanced the malignant progression of GC cells, with increased IL-6 and plasminogen 1 secretion.	[44]
	CXCL2	↑	CXCL2 secreted from omental adipocytes promoted the proliferation and invasion of GC cells through AKT activation.	[45]
	Exosomal miR-155	↑	Exosomal miR-155 secreted by GC cells repressed lipid metabolism and prompted brown adipose differentiation by targeting C/EPBβ of adipose mesenchymal stem cells.	[46]
Vasculature cells				
Endothelial cells	NOVA2	↑	NOVA2 is elevated in cancer endothelial cells of GC, leading to lymph node metastasis.	[51]
	Exosomal miR-214-3p	↑	Exosomal miR-214-3p in vascular endothelial cells suppressed ubiquitin expression editing enzyme A20, which resulted in apatinib resistance.	[52]
	HIPPO, NOTCH, RAS	↑	Tumor endothelial cell clusters contributed to GC growth and metastasis.	[53]
Pericytes	PI3K/AKT, MEK/ERK, BMP pathways	↑	GC exosomes induce pericytes-to-CAFs transition by activating the PI3K/AKT, MEK/ERK, and BMP pathways.	[54]
Innate immune cells				
Macrophage	LOC339059	↓	LOC339059 suppresses M2 macrophage polarization, PDL1 expression, tumor proliferation, and metastasis.	[59]
	COL1A1, COL4A1, COL12A1, PDGFRB	↑	Bioinformatics analyses detected hub genes as prognostic biomarkers correlated with M2 macrophage infiltration in GC.	[60]
TAMs	CXCL8	↑	CXCL8 was secreted under hypoxic environments through the JAK/STAT1 pathway and interaction with the CXCR1/2.	[67]
	GDNF	↑	TAM-derived GDNF enhanced liver metastasis of GC through autophagic activity of GDNF family receptor alpha 1.	[68]
	*MRC1*, *MS4A4A*, *CD36*, *CCL13*, *CCL18*, *CCL23*, *SLC38A6*, *FGL2*, *FN1*, *MAF*	↑	Immunosuppressive macrophage signature showed a highly immunosuppressive phenotype correlated with more persistent tumor activity.	[69]
	*Exosomal* miR-21	↑	M2-polarized TAM-derived exosomes, miR-21, promote the cisplatin resistance of GC cells.	[70]
	*Exosomal* miR-588	↑	M2-polarized TAM-derived exosomes, miR-588, promote the cisplatin resistance of GC cells.	[71]
	LncRNA CRNDE	↑	M2-polarized TAM-derived exosomes, LncRNA CRNDE, promote the cisplatin resistance of GC cells.	[72]
	*Exosomal* miR-223	↑	M2-polarized TAM-derived exosomes, miR-223, promote the doxorubicin resistance of GC cells.	[73]
Neutrophils	CD66-	↑	CD66-positive mature light-density neutrophils were clustering in the peritoneal cavity of patients who underwent laparotomy due to GC.	[79]
	LIF	↑	The development of NETs and peritoneal metastasis of GC enhanced the TGF-β-Smad-LIF axis.	[80]
DCs	ICOS	↑	*H. pylori* infection could increase the expression of ICOS in pDC.	[84,87]
NK cells	*CXCR4*, *RDH8*, *MAGEA11*, *CYP19A1*, *SHOX2*, *GRB14*, *SLC35E4*, *NEK5*, *AKAP5*, *MSI2*, *KYNU*, *PLCL1*	↑	A 12-gene NK cell-associated signature can predict both GC patients’ prognosis and immunotherapy’s efficacy.	[95]
MDSCs	*IL1B*, *CCR1*, *CXCL2*, *GRINA*, *IER3*	↑	Tumor-infiltrating monocytic MDSCs (TI-M-MDSCs) expressed higher levels of genes with immunosuppressive functions.	[98]
	IL-6, IL-8	↑	Serum cytokines IL-6 and IL-8 can promote CD45(+)CD33(low)CD11b(dim) MDSCs to induce arginase I and contribute to CD8^+^ T cell suppression through PI3K-AKT signaling.	[99]
Adaptive immune cells				
T cells	PDCD1, CTLA4, HAVCR2, LAG-3, TIGIT	↓	CD8^+^ T cells exhibited low expression levels of exhaustion markers.	[101]
	CD47, galactin-3	↑	CD47 and galactin-3 were extensively co-expressed in GC cells with peritoneal metastasis and diffuse type.	[102]
	*CXCR4*, *NPC2*, *DDX24*, *ZFP36*, *TGFB1*, *PDCD1*, *NPDC1*, *SRI*	↑	Eight CD8^+^ T cell feature genes generated a novel risk signature for predicting the survival and the effect of immunotherapy in GC patients.	[103]
Treg	Foxp3	↑	The Foxp3 expressing subtype was considered to be related with a worse prognosis due to its function in suppressing CD8^+^ T cells.	[104,105]
	TGF-β	↑	TGF-β prompts Treg differentiation in the GC TME.	[106]
	IDO-1	↑	IDO-1 is up-regulated in *H. pylori*-infected gastric mucosa, enhances differentiation of Tregs, and also decreases the ratio of Th17 cells versus Tregs and the number of Th1 and Th12 cells.	[107]
	TNFR2VEGF	↑	TNFR2-positive Tregs increase with GC malignant progression. TNF-α/TNFR2 pathway prompts Treg activity and induces their immune tolerance.	[108]
B cells	IL-10, TNF-α, VEGF, IL-8, IL-1β	↑	A variety of cytokines, including inflammatory cytokines secreted by a coculture of patient-derived B cells with GC cells, was associated with a worse survival of GC.	[111]
	CXCR4	↑	CXCR4 expressed by tumor-infiltrating B cells revealed a poor prognosis.	[112]
	CXCL13	↑	TLS-associated B cells induce the secretion of CXCL13 and granzyme B in CXCL13^+^CD103^+^CD8^+^ tissue-resident memory T cells via LTα/TNFR2 signaling.	[114]

**Table 2 ijms-25-11735-t002:** Selected ongoing clinical trials of pharmacological strategies targeting TME for GC. GC = gastric cancer, TME = tumor microenvironment, SOX = S-1 and oxaliplatin, PD = programmed cell death, CapeOX = Capecitabine and oxaliplatin, XELOX = Xeloda and oxaliplatin, VEGF = vascular endothelial growth factor, PIGF = placental growth factor, FLOT = 55-Fu, leucovorin, oxaliplatin and docetaxel, CAR T = Chimeric Antigen Receptor T Lymphocytes, TCR = T cell receptor, TIL = tumor-infiltrating lymphocytes, CAR M = Chimeric Antigen Receptor Macrophages, HER2 = human anti-human epidermal growth factor receptor 2, N.A.= not applicable.

Strategy	Agents	Target	Identifier	Patient Number	Status	Results	Comments
Antibody							
	Nivolumab, sintilimab	PD-1	NCT06202781	28	Cohort	Recruiting	SOX + nivolumab or sintilimab
	JS001	PD-1	NCT04744649	N.A.	2 (NICE Trial)	Recruiting	SOX, or XELOX + JS001
	Camrelizumab	PD-1	NCT05545436	34	2	Recruiting	SOX, or XELOX + JS001. Evaluate gene sequence changes (dMMR, MSI, TMB, PD-L1, etc.).
	SHR1210	PD-1	NCT03878472	30	2	Recruiting	S-1 + SHR1210 ± apatinib Evaluate the immunotherapy-related biomarkers, including TME
	Retifanlimab	PD-1	NCT05177133	25	2 (AuspiCiOus)	Recruiting	CapeOX + Retifanlimab Evaluate interferon-gamma expression signature in the TME.
	Nivolumab, Ipilimumab	PD-1, CTLA-4	NCT03342417	60	2	Terminated	Nivolumab + Ipilimumab
	Cadonilimab	PD-1, CTLA-4	NCT06202716	50	2	Not yet recruiting	CapeOX + Cadonilimab for GC with high TMEscore
	Durvalumab	PD-1, CTLA-4	NCT03539822	117	1/2	Active, not recruiting	Durvalumab + cabozantinib ± tremelimumab
	Nivolumab, Relatlimab	PD-1, LAG-3	NCT03044613	32	1	Active, not recruiting	Nivolumab ± Relatlimab before chemoradiation
	Durvalumab	PD-L1	NCT04592913	900	3	Active, not recruiting	Durvalumab + FLOT
	Avelumab	PD-L1	NCT03399071	44	2	Active, not recruiting	Avelumab + FLOT
	Atezolizumab	PD-L1	NCT03448835	20	2 (PANDA)	Unknown status	Atezolizumab + chemothepary
	TTX-030	CD39	NCT04306900	185	1	Completed	TTX-030 + budigalimab + chemotherapy
	BO-112	MDA-5	NCT04508140	18	2a	Terminated	BO-112 + Pembrolizumab for GC patients with liver metastasis.
	PB101	VEGF, PIGF	NCT06075849	30	1	Recruiting	Monotherapy of PB101, an Anti-angiogenic Immunomodulating Agent
	BI-1607	CD32b	NCT05555251		1/2a	Active, not recruiting	BI-1607 + trastuzumab in patients with HER2-positive GC.
	Evorpacept (ALX148)	CD47	NCT05002127	450	2/3 (ASPEN-06)	Recruiting	Evorpacept + trastuzumab, ramucirumab, and paclitaxel in patients with HER2-positive GC.
T cells							
	CAR T cells	CEA	NCT06006390	60	1/2	Recruiting	Intravenous infusion and intraperitoneal injection.
	CAR T cells	CEA	NCT06126406	60	1	Recruiting	Evaluate the safety of CAR-T cell preparations in the treatment of CEA-positive GC.
	CHM-2101	Cadherin 17	NCT06055439	135	1/2	Recruiting	CHM-210 is an analogue cadherin 17 CAR T cell.
	CAR T cells	Claudin18.2	NCT05277987	18	1	Recruiting	Evaluate the tolerability and safety of Claudin18.2 CAR T cell
	CAR T cells	Claudin18.2	NCT06353152	12	1	Recruiting	Evaluate the safety, tolerability, and efficacy of Claudin 18.2- CAR T cell.
	TCR-T cells	K-LC-1	NCT05483491	30	1	Recruiting	Evaluate clinical tumor response to KK-LC-1 TCR-T cell treatment.
	TCR-T cells	K-LC-1	NCT05035407	100	1	Recruiting	Evaluate tolerated dose of KK-LC-1 TCR T cells plus aldesleukin.
	TIL	Natural autologous TIL	NCT06532799	75	1/2	Not yet recruiting	Evaluate the safety profile of TIL in patients with GC.
NK cells	CAR-NK cells	PD-L1	NCT04847466	55	2I	Recruiting	PD-L1 CAR-NK cells +pembrolizumab
	CYNK-101	HER2	NCT05207722	52	1	Active, not recruiting	CYNK-101^+^ Trastuzumab and Pembrolizumab in patients with HER2-positive GC.
Macrophages	CAR M cells	HER2	NCT06224738	9	1	Not yet recruiting	Evaluate the safety and efficacy of HER2 CAR-M in advanced HER2+ GC
Others							
	N.A.	N.A.	NCT056444311	N.A.	Cohort	Recruiting	Assess the immune tumor environment profiles and MRD using liquid biopsies through neoadjuvant chemotherapy.

## Data Availability

Not applicable.

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
