# Peer review of "Molecular Mechanism for Malignant Progression of Gastric Cancer Within the Tumor Microenvironment"

_ijms, 2024, doi:10.3390/ijms252111735_

Round 1

Reviewer 1 Report

Comments and Suggestions for Authors

Dear authors of “Molecular Mechanism for Malignant Progression of Gastric Cancer within the Tumor Microenvironment”,

Thanks for giving me the opportunity to review your manuscript. This is an interesting and rich state of art on the multiple roles of the tumor microenvironment (TME) in cancer and more specifically in gastric cancer. The manuscript is well written and organized. It is included updated on recent articles.

Many aspects are covered, such as the different types of cells, the different signaling pathways, and the consequences on TME composition. Those elements are highlighted separately, which helps the reader to follow the review and makes the manuscript quite attractive. Impacts on tumor malignancy, such as cell proliferation, migration, or invasion, are also present, as is the involvement of the gut microbiote.

Raised comments after previewing this manuscript have been considered since the first submission was focused only on immune cells. Now, this review is not anymore only on the tumor immune microenvironment, which highly improved its impact within the scientific community.

In conclusion, the manuscript is still well written, organized, and suitable for publication.

Regards

Author Response

Comments 1: Dear authors of “Molecular Mechanism for Malignant Progression of Gastric Cancer within the Tumor Microenvironment”, Thanks for giving me the opportunity to review your manuscript. This is an interesting and rich state of art on the multiple roles of the tumor microenvironment (TME) in cancer and more specifically in gastric cancer. The manuscript is well written and organized. It is included updated on recent articles. Many aspects are covered, such as the different types of cells, the different signaling pathways, and the consequences on TME composition. Those elements are highlighted separately, which helps the reader to follow the review and makes the manuscript quite attractive. Impacts on tumor malignancy, such as cell proliferation, migration, or invasion, are also present, as is the involvement of the gut microbiote. Raised comments after previewing this manuscript have been considered since the first submission was focused only on immune cells. Now, this review is not anymore only on the tumor immune microenvironment, which highly improved its impact within the scientific community. In conclusion, the manuscript is still well written, organized, and suitable for publication.

Regards

Response 1: We appreciate the reader’s comments. We fully agree with what you pointed out. We would like to make these comments useful for my work in the future.

Reviewer 2 Report

Comments and Suggestions for Authors

In this manuscript, the authors reviewed the roles of various cell components of TME in the regulation of the malignant progression and metastasis of GC and analysed how signaling pathways play an important part in mediating the interaction between cancer cells and the different components of the GC TME.

Overall, the manuscript is of interest. However, several issues needs to be addressed:

1)       Figure 2 need to be revised: in particular, are you sure that TAM derived from M2 macrophages? What about M1-like TAMs population?

2)       Line 267-272 need to be revised. The authors need to explain the presence of M1-like TAMs in tumor tissues.

3)       Figure 3 shows the schematic representation of the main signaling pathways that are activated in tumor cells or tumor-associated stromal cells. Could be better distinguish the signaling pathways between cancer cells from stroma/immune cells of TME?

4)       Figure 3: among receptors represented, CSF-1R is missing! On the base of recent reports (PMID: 38254773 and others), the authors should revise the text related to paragraph 3.

5)       Acronyms need to be revised.

6)       Organization of sections should be revised.

7)       Section 4: microbiota and TME is of interest and need to be extended.

Comments on the Quality of English Language

Extensive revision of English editing is required.

Author Response

In this manuscript, the authors reviewed the roles of various cell components of TME in the regulation of the malignant progression and metastasis of GC and analysed how signaling pathways play an important part in mediating the interaction between cancer cells and the different components of the GC TME.

Overall, the manuscript is of interest. However, several issues needs to be addressed:

Comments 1; Figure 2 need to be revised: in particular, are you sure that TAM derived from M2 macrophages? What about M1-like TAMs population?

Response 1: Thank you for the reviewer’s helpful comments.  As the reviewer recommended, we changed Figure 2 and add an M1-like TAM population.

Comments 2; Line 267-272 need to be revised. The authors need to explain the presence of M1-like TAMs in tumor tissues.

Response 2: We appreciate the reviewer's comments. In compliance with the reviewer’s suggestion, we added a paragraph on the presence of M1-like TAMs in the text on rows 299-303.

Comments ;3 Figure 3 shows the schematic representation of the main signaling pathways that are activated in tumor cells or tumor-associated stromal cells. Could be better distinguish the signaling pathways between cancer cells from stroma/immune cells of TME? 

Response 3: We appreciate the reviewer's comments. According to the reviewer’s suggestion, we changed Figure 3 to distinguish the signaling pathways between cancer cells from stroma/immune cells of TME.

Comments 4; Figure 3: among receptors represented, CSF-1R is missing! On the base of recent reports (PMID: 38254773 and others), the authors should revise the text related to paragraph 3.

Response 4: Thank you for the reviewer’s helpful comments. As the reviewer recommended, we added a paragraph on the role of CSF-1R in GC TME in the text on rows 6619-632 and in Figure 3.

Comments 5; Acronyms need to be revised.

Response 5: We revised acronyms, such as Helicobacter pylori, gastric cancer, and natural killer, connective tissue growth factor carefully. Significant changes were highlighted in yellow.

Comments 6; Organization of sections should be revised.

Response 6: We appreciate the reviewer’s comments. In compliance with the reviewer’s suggestion, we added the new section “TME-based cancer therapy in GC” on rows 740-8820.

Comments 7; Section 4: microbiota and TME is of interest and need to be extended

Response 7: According to the reviewer’s suggestion, we added a paragraph on the role of gut microbiota in GC TME in the text on rows 706-718, 730-735, and 801-820.

Reviewer 3 Report

Comments and Suggestions for Authors

This manuscript analyzes in great detail the tumor microenvironment (TME) of gastric cancer (GC)

I think this work is well organized, presented and written.  The authors listed the cellular components (stromal cells and immune cells) as well as the pathway involved in the proliferation of GC. Also, they analyzed the possible targets of therapy. Some new findings have been reported regarding the new means (spheroids and organoids) used to study GC.

Overall, the description is detailed and up-to-date.

I would suggest a short focus on molecular mechanisms for activation of the immune system against GC (activating receptors involved or relevant in the context). Indeed, the authors talk about the mechanisms of inhibition of the immune response, but activation should be present to favor the blocking of GC growth.

For instance, some papers have been published on NKG2D in relevant scientific journals that merit mention. The same is true for other activating receptors.

Also, the role of ADAMs should be considered.

Anyway, the manuscript is a good one and certainly it will be improved with the indicated additions.

Comments on the Quality of English Language

English is good

Author Response

This manuscript analyzes in great detail the tumor microenvironment (TME) of gastric cancer (GC)

I think this work is well organized, presented and written.  The authors listed the cellular components (stromal cells and immune cells) as well as the pathway involved in the proliferation of GC. Also, they analyzed the possible targets of therapy. Some new findings have been reported regarding the new means (spheroids and organoids) used to study GC.

Overall, the description is detailed and up-to-date.

Comments 1: I would suggest a short focus on molecular mechanisms for activation of the immune system against GC (activating receptors involved or relevant in the context). Indeed, the authors talk about the mechanisms of inhibition of the immune response, but activation should be present to favor the blocking of GC growth.

For instance, some papers have been published on NKG2D in relevant scientific journals that merit mention. The same is true for other activating receptors.

Also, the role of ADAMs should be considered.

Response 1: We appreciate the reviewer’s comments. In compliance with the reviewer’s suggestion, we added a paragraph on the role of immune activating receptors in GC TME in this review in the Discussion section on rows 871-910. We also mention about ADAMs on rows 890-900.

Round 2

Reviewer 2 Report

Comments and Suggestions for Authors

The authors addressed all the raised questions. The manuscript is improved and is suitable for publication.